# Investigation of the In Vitro and In Vivo Biocompatibility of a Three-Dimensional Printed Thermoplastic Polyurethane/Polylactic Acid Blend for the Development of Tracheal Scaffolds

**DOI:** 10.3390/bioengineering10040394

**Published:** 2023-03-23

**Authors:** Asmak Abdul Samat, Zuratul Ain Abdul Hamid, Mariatti Jaafar, Chern Chung Ong, Badrul Hisham Yahaya

**Affiliations:** 1Lung Stem Cell and Gene Therapy Group, Department of Biomedical Sciences, Advanced Medical and Dental Institute, Universiti Sains Malaysia, Sains@Bertam, Kepala Batas 13200, Malaysia; 2Department of Fundamental Dental and Medical Sciences, Kulliyyah of Dentistry, International Islamic University Malaysia, Kuantan 25200, Malaysia; 3School of Materials and Mineral Resources Engineering, Universiti Sains Malaysia, Nibong Tebal 14300, Malaysia; 4Fabbxible Technology, 11a Jalan IKS Bukit Tengah, Tmn IKS Bukit Tengah, Bukit Mertajam 14000, Malaysia

**Keywords:** biocompatibility, thermoplastic polyurethane, polylactic acid, degradation, pH, inflammatory response

## Abstract

Tissue-engineered polymeric implants are preferable because they do not cause a significant inflammatory reaction in the surrounding tissue. Three-dimensional (3D) technology can be used to fabricate a customised scaffold, which is critical for implantation. This study aimed to investigate the biocompatibility of a mixture of thermoplastic polyurethane (TPU) and polylactic acid (PLA) and the effects of their extract in cell cultures and in animal models as potential tracheal replacement materials. The morphology of the 3D-printed scaffolds was investigated using scanning electron microscopy (SEM), while the degradability, pH, and effects of the 3D-printed TPU/PLA scaffolds and their extracts were investigated in cell culture studies. In addition, subcutaneous implantation of 3D-printed scaffold was performed to evaluate the biocompatibility of the scaffold in a rat model at different time points. A histopathological examination was performed to investigate the local inflammatory response and angiogenesis. The in vitro results showed that the composite and its extract were not toxic. Similarly, the pH of the extracts did not inhibit cell proliferation and migration. The analysis of biocompatibility of the scaffolds from the in vivo results suggests that porous TPU/PLA scaffolds may facilitate cell adhesion, migration, and proliferation and promote angiogenesis in host cells. The current results suggest that with 3D printing technology, TPU and PLA could be used as materials to construct scaffolds with suitable properties and provide a solution to the challenges of tracheal transplantation.

## 1. Introduction

The treatment for extensive tracheal injuries is complex. For almost a century, clinicians and researchers have attempted to develop a graft to replace long-segment tracheal defects. The tracheal replacements used clinically range from autologous tissue flaps and patches, allograft transplants, synthetic stents, and prostheses to tissue-engineered scaffolds [1,2]. Advances in tissue engineering provide promising alternative approaches for assembling functional constructs that repair, preserve, or enhance defective tissues or organs [3,4]. Biodegradable polymers are gaining popularity in tracheal tissue engineering, particularly for paediatric patients, due to the limited treatment options available to children compared to adults [5,6]. Apart from providing mechanical support for the injured trachea, the scaffold should facilitate cellular migration and proliferation, tissue modification, and degradation at an appropriate rate during growth to eliminate recurrent surgeries [7,8]. Although synthetic scaffolds demonstrate promise for future applications, biocompatibility, graft mobility, and poor integration with the host tissue are concerns that must be addressed. Additionally, biodegradable scaffolds introduce additional challenges, such as releasing toxic degradation products over time [7].

Thermoplastic polyurethanes (TPU) are synthetic polymers commonly derived from petrochemical-based polyol [9], joined with a diisocyanate and a chain extender by ring-opening polymerisation to form linear, uncrosslinked, segmented copolymers consisting of alternating hard and soft segments. The soft and flexible parts are derived from polyols such as polyester or polyether, while the rigid and hard segments are form from the diisocyanate and chain extender [10]. TPUs are polymeric materials that can be manipulated, moulded, and produced through heating in various industrial processes. TPU exhibits a broad range of mechanical properties across various temperatures due to the different ratios of soft to hard segments. As a result of its excellent physical properties and biocompatibility, it is widely used in biomedical applications, particularly in flexible uses such as blood vessels [11,12,13], catheters [14,15], and cartilage [16,17]. Polylactic acid (PLA) is a semi-crystalline polymer that belongs to the α-hydroxy acid family, derived from renewable sources such as corn, potatoes, sugarcane, and beets [18,19]. It is classified as an aliphatic polyester because of the ester bonds that connect the monomer units, the lactic acids [20,21]. Lactic acid is a critical component of the glycolytic energy cycle in organisms and is necessary for the growth and development of living organisms [22]. Therefore, PLA and its copolymers have become one of the most researched components in the biomedical field because of their excellent biological and mechanical properties, biodegradability, and processability. PLA products were approved by the US Food and Drug Administration (FDA) for direct contact with biological fluids in 1970 [22,23]. Hence, it is present in a wide range of applications such as medical implants, sutures [24], bone fixation screws [25], and drug delivery systems [26]. However, biodegradable PLA exhibits little to no elastic behaviour and is not favoured for applications requiring high flexibility or deformation in situ [23]. A few studies have demonstrated the feasibility of combining TPU and PLA using different methods such as microcellular injection moulding [27], thermally induced phase separation [28], and 3D printing [29]. It is hypothesised that combining the two materials and the subsequent 3D printing method produces a composite with suitable qualities for tracheal replacement.

The advent of additive manufacturing (AM) as a new production process has triggered a massive change in the fields of manufacturing, engineering, aerospace, and medicine [29]. Three-dimensional (3D) scaffolds fabricated using AM, also known as rapid prototyping (RP), are a promising strategy in tissue engineering for the replacement and regeneration of damaged tissues [30]. The key features of 3D printing are the elimination of constraints on the design and the production of intricate geometries using the least amount of material. The technique involves making three-dimensional objects from 3D-modelled data in a progressive layer-wise deposition using printing technologies and is a potential tool for producing scaffolds for personalised treatments [31,32]. It has been applied in regenerative medicine to manufacture bone grafts [33,34,35], trachea [36], meniscus [37], and cartilage [38].

As the coronavirus disease 2019 (known as COVID-19) pandemic severely impacts respiratory function, it has put tremendous strain on global ventilator supply chains. An increase in supply is urgently needed as hospitals are overcrowded with patients requiring comprehensive respiratory care [39]. According to the Centers for Disease Control and Prevention, up to 6% of patients need to be admitted to intensive care units (ICUs) and require hospitalisation, with mechanical ventilation being the most common requirement [40]. In these circumstances, 3D printing is advantageous for prostheses and medical implants. Patients can recover more quickly from surgery and have a higher success rate when implants are custom-made. Smaller production runs are also more cost-effective if they are printed immediately and on-site. This strategy has a distinct advantage in that open-source ideas can be disseminated worldwide [41].

Biocompatibility is the most important criterion for a biomaterial to succeed as a medical device for an implant. According to Williams (2008), ‘The biocompatibility of a scaffold or matrix for a tissue engineering product refers to the ability to perform as a substrate that will support the appropriate cellular activity, including the facilitation of molecular and mechanical signalling systems to optimise tissue regeneration, without eliciting any undesirable local or systemic responses in the eventual host’ [42]. A biomaterial scaffold intended for implantation should not be carcinogenic, immunogenic, or toxic to living tissue [43,44,45]. Based on recommendations by the international standard ISO 10993 (Biological Evaluation of Medical Devices), all materials intended for use in humans should be subjected to in vitro and in vivo biocompatibility tests to assess the response and behaviour of cells interacting with them [46,47].

In this study, 3D-printed scaffold discs made of TPU and PLA polymers were developed and their biocompatibility was evaluated using in vitro and in vivo studies. As a viable material for tracheal tissue engineering, the TPU/PLA blended matrix offers good biocompatibility for cells and tissues, and 3D printing may be one of the best options to fabricate not only tracheae but also other biomedical devices.

## 2. Materials and Methods

### 2.1. Materials

TPU Estane 58311 NAT 028 (Brussel, Belgium) and PLA NatureWorks^®^, 2003D were purchased from NatureWorks LLC, Plymouth, MI, USA, with a specific gravity of 1.24 and melt index of 5.0–7.0 g/10 min (2.16 kg loads at 210 °C). The cell line normal human bronchial epithelial (BEAS-2B) cells were purchased from American Type Culture Collection (ATCC) (Manassas, VA, USA). BEAS-2B is the most common cell line used to evaluate tracheal scaffolds in vitro [48,49,50]. The cells were cultured in alpha minimum essential medium (α-MEM) (Invitrogen, Carlsbad, CA, USA) supplemented with 10% foetal bovine serum (FBS) (Thermo Fischer Scientific, Waltham, MA USA) and 1% penicillin/streptomycin (Sangon, Shanghai, China). Cell growth was measured using a 3-(4,5-dimethylthiazol-2-yl)-2,5-diphenyl-tetrazolium bromide test, also known as MTT assay (Sigma Aldrich, St. Louis, MI, USA). Other materials used were Hoechst 33342 (Thermo Fisher Scientific, Waltham, MA, USA), KaryoMAX^®^ Colcemid^TM^ (Thermo Fischer Scientific, Waltham, MA USA), ketamine/xylazine cocktail (Troy Laboratories Pty Limited, Glendenning, Australia), xylene (Bendosen), Harris haematoxylin (Sigma-Aldrich, Gillingham, UK), ammonia water (Merck), eosin (Sigma-Aldrich, Gillingham, UK), and hexamethyldisilazane (HMDS) (Sigma-Aldrich, Gillingham, UK).

### 2.2. 3D Printing of the TPU/PLA Scaffolds

The TPU/PLA filament feedstock (ratio 90/10) was produced according to the method used in our previous study [51]. The TPU and PLA pellets were dried in a 60 °C oven for 12 h. Then both materials were manually premixed in a plastic zip-lock bag and fed into the hopper of a Brabender (Duisburg, Germany) single screw extruder according to the manufacturer’s instructions. The rotation speed was set to 40 (±5) rpm, while the temperatures of the heating zones ranged from 170 °C to 195 °C, and a 1.75 mm die head was used. The extruded filament was quenched in a water bath and manually pulled to form a filament with a constant diameter ranging from 1.65 mm to 1.85 mm.

3D printed circular discs were designed utilising computer-aided SolidWorks 2017 software version 25 (Boston, Massachusetts, USA). The designed files were converted to STL (standard tessellation language) format and imported into an open source slicing application, Cura Ultimaker 4.7 software (Zaltbommel, The Netherlands), generating a printer-specific G-code which instructed the 3D printer model Artillery Sidewinder X1 (Shenzhen, China) during the printing process. The temperature of the nozzle extruder used was 200 °C whilst the speed was set to 25, 15, and 30 mm/sec for pure TPU, TPU/PLA, and pure PLA respectively. The diameter of the nozzle or layer thickness used in this printer was 0.2 mm, and the infill density was set to 70%.

### 2.3. In Vitro Study

Before in vitro testing, TPU/PLA pellets were sterilised by immersing them in 70% ethanol (*v/v*) for 2 h, followed by rinsing three times with 1x phosphate buffered solution (PBS) to eliminate all traces of ethanol. The pellets were then air-dried in a sterile atmosphere before being sterilised for 2 h with ultraviolet light. This phase ensured that any pollutants on the surface of the pellets were removed.

#### 2.3.1. In Vitro Degradation Study

The in vitro degradation test was conducted according to the ASTM F1635-11 standard, designed to determine the degradation rates of polymers and devices made from resorbable polymers. The samples were weighed after drying overnight at 60 °C. Each sample was individually enclosed in a plastic container filled with a 1xPBS solution and incubated at 37 °C on an orbital shaker, Stuart S1500 (Illinois City, IL, USA), at a shaking rate of 50 rpm. Every seven days, PBS was refreshed, and the test lasted up to 8 weeks. The samples were rinsed three times with purified water before being dried overnight to achieve constant weight, and weighed at each time point. Equation (1) was used to measure the weight loss of the materials:(1)$$Degradation\;\left(\%\right)=\left(1-\frac{W_{n}}{W_{o}}\;\right)\times 100$$

*W_0_* is the initial sample weight, and *W_n_* is the weight of the same sample after degradation for a time, *n*. Three samples were used for each composition in this experiment.

##### 2.3.2. pH Analysis of Degradation Extract

pH changes in the medium were measured by immersing the TPU, PLA, and TPU/PLA blend samples for eight weeks in PBS at pH 7.4, whilst PBS was used as the control. In addition, the pH of the degradation medium was measured with a pH metre, Model H1 2213 pH/ORP HANNA Instrument (Woonsocket, Rhode Island, USA) every week. For each composition, three samples were prepared, and the average pH of the three samples was reported.

#### 2.3.3. Preparation of 3D-Printed Scaffold Extracts

For quantification of the cell proliferation assay and the scratch assay in the following subsection, extracts of the TPU, TPU/PLA, and PLA scaffolds were used according to the ISO 10993-12:2021 [52]. The extracts were prepared by immersing the scaffolds for 24 h in complete α-MEM and in low serum α-MEM for proliferation and scratch assays, respectively. The extracts were kept in a 4 °C refrigerator and centrifuged to separate the degradation particles before being used in the assays.

#### 2.3.4. MTT Cell Proliferation Assay

Due to the risk of damaging the scaffold structure when using the direct technique, the indirect technique was adopted to evaluate proliferative cells. In addition, the number of formazan crystals produced, debris, and precipitated proteins can interfere with the assay’s optical measurements [53]. Cell growth was measured using the 3-(4,5-dimethylthiazol-2-yl)-2,5-diphenyl-tetrazolium bromide test, also known as the MTT assay (Sigma Aldrich, St. Louis, MI, USA). The assay relies on metabolically active cells that convert the MTT tetrazolium salt to purple formazan. This test identifies colorimetric variations in the number of live cells and their metabolic activity, which can be quantified using spectrophotometry.

BEAS-2B cells were grown to confluence in complete α-MEM containing 10% FBS and 1% antibiotic–antimycotic (AA) solution at 37 °C in an incubator with 5% carbon dioxide (CO_2_). In 24-well plates, 500 BEAS-2B cells were seeded per well and cultured for seven days. The extract media were removed after one, three, and seven days and fresh medium containing the MTT solution (5 g/L) was applied to each well, followed by a 4-hour incubation. MTT is light sensitive, so the plates were covered with aluminium foil. After 4 h, the medium was withdrawn, and 1 mL dimethyl sulfoxide (DMSO) was added for 15 min to dissolve the formazan. After dissolving all formazan crystals, 100 µL aliquots were transferred to a 96-well plate, with five replicates per sample. Absorbance was measured in triplicate using a microplate spectrophotometer (Bio-Tek Instruments, Winooski, VT, USA) at a wavelength of 570 nm, with DMSO serving as the blank.

#### 2.3.5. Scratch Assay

An in vitro scratch or wound-healing assay was performed to measure the unidirectional migration of BEAS-2B cells and investigate the effects of TPU, TPU/PLA, and PLA extracts on cell migration [54]. The scratch assay is a well-established in vitro method of creating a cell-free region on a monolayer of cells to mimic a wound, which is mostly used to observe collective cell migration in two dimensions [55]. BEAS-2B cells were seeded at a density of 1.5 × 10^5^ cells/well in a 12-well plate and cultured in complete α-MEM until 100 per cent confluence. Once the cells were almost 100 percent confluent, they were treated for 2 h with 10 g/mL KaryoMAX^®^ ColcemidTM (Gibco, NY, USA) to facilitate cell synchronisation. Next, a scratch was made in the confluent layer across the diameter of the plate using a 200 µL pipette tip. After that, the cells were rinsed with PBS to eliminate any free-floating cells or debris. The extract medium from each scaffold was added to the well, whereas the control well contained low serum alpha-MEM. The plate was incubated for 24 h at 37 °C with 5% CO_2_. Phase-contrast pictures were acquired at 0, 24, and 48 h of incubation using an inverted microscope (Olympus IX-71), (Tokyo, Japan). The percentage of migrating areas in all photos was analysed and measured with ImageJ 1.38e software (NIH, Bethesda, MD, USA) using three replicates for quantification.

#### 2.3.6. Cell Attachment Assay

A cell attachment assay was conducted to analyse the ability of the cells to attach to the 3D-printed scaffolds. In this assay, 1.2 × 10^5^ BEAS-2B cells were seeded directly onto sterile scaffolds in complete α-MEM for 3, 7, and 10 days using the saturation of cell suspension technique [56]. Half of the total cell suspension volume was initially added to the top of each scaffold. Next, the prepared plate was gently transferred to an incubator at 37 °C, with 95% humidity and 5% CO_2_. After an hour, the remainder of the medium was added to the scaffolds and incubated for 30 min. Then, the scaffolds were placed in different sterile tissue culture wells, one scaffold per well. Finally, fresh medium was added to the plates and changed every three days. The scaffolds were fixed for Hoechst 33342 staining and visualised using an Olympus IX70 inverted fluorescence microscope (Tokyo, Japan). The assay was performed using three samples for each composition.

### 2.4. In Vivo Study

#### 2.4.1. Source of Animals and Ethical Approval

Twenty-five male (n = 25) Sprague Dawley rats with an average weight of 316.6 ± 42.0 g and ages of 6–8 weeks were acquired from the Animal Research and Service Centre (ARASC), Universiti Sains Malaysia. All the animals were subjected to surgical procedures according to the approval of the Universiti Sains Malaysia Animal Ethics Committee. The rats were divided randomly into five groups. Each animal that received a 3D printed scaffold had two scaffolds of the same type placed on the dorsal parts’ left and right pockets. The sham group underwent surgery without a scaffold to imitate an injury in the animal. Finally, animals without surgery or implantation were in the naive group. The animals were housed in a polypropylene cage (425 × 266 × 185 mm) (Techniplast, Buguggiate, Italy) with wood shavings (Whitten Molen, Meeuwen, The Netherlands). The lighting was controlled on a 12-hour light and 12-hour dark cycle. The animals had access to food and water ad libitium.

#### 2.4.2. Subcutaneous Implantation of 3D Printed Scaffolds

The rats were anaesthetised using an intraperitoneal injection of ketamine/xylazine cocktail (Troy Laboratories Pty Limited, Glendenning, Australia) at 0.2 mL/100 g. The dorsal area was shaved and scrubbed with chlorhexidine, 70% ethanol and povidone iodine. An incision was made at 2 cm from the midline of the dorsum, and a subcutaneous pocket was created. The scaffold was inserted and sutured using an absorbable suture. Each animal received a subcutaneous injection of 1 mg/kg analgesic meloxicam administered immediately after surgery and 24 h later to avoid post-operative pain. The animal was placed in the recovery position in a clean cage with a hot pack and towel to prevent hypothermia. The heating pad was removed once the animal started to move. The post-operative care was carried out immediately, and the animals were housed in one cage per animal to allow proper healing. The animals were monitored and examined daily for any bleeding, infection, or behavioural changes.

#### 2.4.3. Histological Assessment of Explanted Skin Tissues and Other Organs

(a)Post-mortem tissue handling

Animal were euthanised with 10% CO_2_ for five minutes at each time point. They were confirmed dead when they stopped breathing, did not have a heartbeat, and did not respond to a firm toe pinch. The animals were shaved, a dorsal midline incision was made using a scalpel, and the adjacent fascia was released via gentle dissection to find the explanted tissue. The dissected tissues were transferred into a bowl filled with 1xPBS before being fixed in the 10% neutral buffered formalin.

(b)Tissue processing and embedding

The skins with implanted scaffolds, kidneys, and livers of the euthanised animals were fixed in 10% (*v/v*) neutral buffered formalin overnight and processed and processed in an automated Excelsior™ AS Tissue Processor (Thermo Fisher Scientific, Waltham, MA, USA) for 16 h. Tissues were embedded in paraffin (Paraplast^®^ Plus, Leica Biosystems, Richmond, IL, USA) and cut into 3 to 5 μm sections with a rotary microtome Histo-Tek SRM II (Sakura Finetek, Torrance, CA, USA). The sections were transferred into a water bath at 37 °C and then mounted on frosted-end slides (HmbG GmbH, Hamburg, Germany) for basic staining. 

#### 2.4.4. Haematoxylin and Eosin (H&E) Staining

The sections were subjected to standard H&E staining (Sigma-Aldrich, St. Louis, MI, USA) to examine histopathological changes in the skin. The process included deparaffinisation with xylene (Merck, Lowe, NJ, USA), and rehydration with graded ethanol (100%, 90%, 80%, and 70%), and tap water, each for 2 min. It was followed by staining with Harris haematoxylin (Sigma-Aldrich, St. Louis, MI, USA) for 11 min, 0.5% acid alcohol for 2 s, tap water for 2 min, 0.2% ammonia water (Merck, Lowe, NJ, USA) for 20 s, and tap water for 2 min. Then the samples were counter-stained with eosin (Sigma-Aldrich, St. Louis, MI, USA) for 3 min. Next, the dehydration process was started using graded ethanol (70%, 80%, 90%, and 100%) for 10 s, followed by clearing using xylene for 2 min.

#### 2.4.5. Scanning Electron Microscopy for Tissue Samples

All the tissues for SEM were fixed in McDowell–Trumps fixative at 4 °C overnight. The McDowell–Trumps was prepared using a mixture of PBS, glutaraldehyde (Sigma-Aldrich, St. Louis, MI, USA), formaldehyde (Sigma-Aldrich, St. Louis, MI, USA), and distilled water. The samples were washed in PBS, fixed in 1% osmium tetroxide (OsO_4_) (Crescent Chemical Co. Inc., Islandia, NY, USA), and distilled water. Next, the samples were dehydrated through a graded series of ethanol with 50%, 70%, 95%, and 100% concentrations, followed by hexamethyldisilazane (HMDS) (Sigma-Aldrich, Gillingham, UK) for 15 min. The samples were allowed to dry in a desiccator overnight. Finally, all samples were coated with gold before imaging using a scanning electron microscope (Hitachi SEM S4700) (Tokyo, Japan).

### 2.5. Morphometric Analyses

To evaluate the inflammatory response and angiogenesis associated with subcutaneous implantation of 3D-printed scaffolds, morphometric analyses were performed. Briefly, H&E stained slides (n = 2) of each group were analysed for inflammatory cell infiltration or the number of blood vessels in a given area, blinded to avoid bias in the evaluation, and reviewed by a pathologist. Blinding was achieved by labelling individual samples without reference to a group, and minimal background information was provided. For each slide, a total of fifteen measurements were taken randomly at five different locations. Three slides were analysed per animal, with two animals per group. The total number of inflammatory cells and blood vessels in the field was calculated at 200× magnification.

### 2.6. Statistical Analysis

Data are represented as mean ± SD. Data were subjected to analysis of variance (ANOVA); one-way ANOVA, followed by *post hoc* Tukey’s test, and two-way ANOVA, depending on the number of variables analysed. The first statistical test was applied when only one variable was considered. In contrast, the second method was used in cases wherein more than one variable was considered. Significance levels were set at *p*  <  0.05. GraphPad Prism (San Diego, CA, USA) software was used for statistical analysis and graphing.

## 3. Results and Discussion

### 3.1. Morphology and Properties of the Scaffolds

The 3D-printed scaffolds were produced using the FDM method. The scaffolds were printed with a porous and lattice-like structure that starts at the bottom and outermost edges and continues upwards. In each layer, the filament was stacked in a line pattern, while the next layer was stacked at a 90-degree angle to the first. The TPU and TPU/PLA composites were comparable in colour and texture, but the PLA scaffolds were whiter, more transparent, and had a smoother surface. The TPU/PLA combination, on the other hand, was much less transparent and had a yellowish hue and a rougher surface. Cross-sections of the scaffolds were examined with an SEM at 100× magnification and showed that the surface of each scaffold was smooth. In addition, the filament thickness of PLA appeared to be more uniform than that of TPU and TPU/PLA blends. The fibres of the PLA scaffolds were well-defined and more evenly distributed. In contrast, the fibres of the TPU/PLA scaffolds were rougher and more randomly distributed, with the TPU fibres appearing to be the most irregular. Photographic images of the 3D-printed scaffold discs and SEM images of the filaments are shown in Figure 1, while Table 1 shows the mean fibre diameter, mean pore size, and porosity for each polymer type.

FDM is the most popular 3D printing technology due to its simple concept, which does not involve hazardous solvents [57,58], and, most importantly, the printing apparatus is affordable and easily fits on a tabletop [59]. In this study, no significant differences were observed in the morphology of the 3D printed scaffolds between the TPU, PLA, and TPU/PLA blend. This result is in line with the report by Heidari-Rarani et al. (2020) and Auffray, Gouge, & Hattali (2022), which demonstrated that with an optimal setting of the parameters of the 3D printer using FDM, the preferable design can be controlled [29,60,61]. Successful tissue regeneration requires scaffolds with particular mechanical stability or biodegradability, appropriate size, and porosity to provide a suitable microenvironment for sufficient cell–cell interaction, cell migration, proliferation, and differentiation. Hence, FDM becomes a potential method to be employed in tissue engineering at a short lead time and low manufacturing cost. Pore size and distribution in scaffolds play an essential role in the growth of cells and the regeneration of tissues. The 3D porous structure allows infiltration, adhesion, and proliferation of cells, thus promoting nutritional and metabolic waste exchange and stimulating angiogenesis [62,63]. Nonetheless, the porosity should be sufficiently small to facilitate mechanical interlocking between the cellular tissue and the three-dimensional scaffold. In addition, porosity must provide structurally stable support for the weight of the tissue and be large enough to effectively transport cellular waste and nutrients for cell growth [64,65]. Without adequate pore size, distribution, or connectivity, there is a risk of cell mortality due to starvation or insufficient cell dispersion to the centre of the scaffold [66].

### 3.2. Degradation and pH Analysis

During the eight-week incubation period, there was no degradation of the PLA scaffold. In contrast, TPU disintegrated at the fastest rate with up to 10 percent, followed by the TPU/PLA scaffold combination with 5 percent. The longer the incubation time, the more the TPU and the composite degraded, as shown in Figure 2a. The effect of material types and time on the rate of degradation was analysed using two-way ANOVA. It was detected that the material (pure TPU and blended TPU/PLA scaffolds) as well as time had a significant effect on the degradation rate (*p*-value < 0.001).

The pH of PLA was almost similar to that of the control group throughout the experiment, but TPU and TPU/PLA scaffolds showed a gradual decrease in pH to below 7, as shown in Figure 2b. The results show that the PLA degradation extract has no significant effect compared to the control group and maintains a relatively constant pH. On the other hand, TPU and its composite scaffold showed a significant decrease compared to the control and PLA groups (*p*-value < 0.001). Furthermore, significant differences in the degradation extracts of TPU and its composite were observed with longer incubation time, but not with PLA.

The cytotoxicity of the degradation products is crucial for the development of biodegradable and biocompatible polymers. Implantation of these polymers may release tiny particles that alter the pH. The by-products of the polymers should be non-toxic and harmlessly excreted through the body system [67,68]. The results show that the pH gradually decreases over time for TPU and TPU/PLA blends, but not for PLA; the lowest pH measured after eight weeks was slightly below 7. The result correlates with our findings on degradation, which showed that TPU and its blend degrade faster than PLA. This result is consistent with a study by Gao et al. (2019), which reported that the mass loss of 3D-printed scaffolds was 1.1% after two weeks and 2.5% after four weeks, providing further evidence that the degradation rate of PLA scaffolds is indeed slow [69]. However, when PLA was produced using the thermally-induced phase separation method, PLA showed a dramatic weight loss of more than 40% within four weeks, as reported by Jing et al. (2014) [28]. In contrast, TPU and TPU/PLA blends prepared using this method, showed a prolonged degradation rate.

The biodegradation of TPU and PLA mainly depends on hydrolytic degradation of polymer chains, classified into two types: surface erosion and bulk erosion. Surface erosion occurs exclusively at the polymer–water interface, while bulk erosion occurs uniformly throughout the polymer [70]. Degradation of polyurethane occurs when water molecules infiltrate the polymer network, triggering hydrolysis of the polyurethane chains, including the chemical dissolution of ester and amide bonds. Similarly, the hydrolytic degradation of PLA starts by breaking the ester link of the polymeric chain [70,71]. Carbon dioxide is one of the components generated during the hydrolytic decomposition of TPU [72,73], and hence we observed a pH decrease. Another possible explanation is that the breakdown product of PLA was lactic acid; however, because the degradation rate of PLA was consistent in our investigation and the pH generally did not change throughout the experiment, this is not a plausible explanation.

### 3.3. Proliferation and Wound Healing Effects

This study also investigated the effects of the TPU/PLA mixture and extracts on BEAS-2B cell proliferation ability and migration ability, whether they promote or inhibit cell–cell interaction in a scratch assay. The morphology of proliferating cells in the experimental groups appeared normal and comparable to the control group, as shown in Figure 3. Consistent growth was evident in the control group, and all groups reached almost similar confluence after seven days. Furthermore, the MTT results again showed a significant increase in the metabolic activity of the cells cultured in the extract media, with the type of material significantly (*p*-value < 0.001) affecting the metabolic activity of the cells, and none of the scaffolds being toxic to the cells.

The wound healing scratch test can be used to determine whether the scaffold or its degradation products accelerate or delay wound healing in vitro [55]. The results obtained with the scratch test are shown in Figure 4. Overall, it was visible that TPU, TPU/PLA, and PLA extracts did not inhibit cell migration in the in vitro wound model compared to the control group. There were no significant differences (*p*-value > 0.05) between the material types in terms of cell migration rate. After 24 h of treatment, BEAS-2B cells had covered 50% of the wound area and almost completely closed the site within 48 h.

Cell proliferation and migration are required for physiological and pathological processes such as wound healing, revascularisation, and tissue regeneration. Cell migration on scaffolds is critical for tissue regeneration as it closely mimics cell interaction during wound healing in vivo. In this experiment, the cells showed similar growth to the control group and were not affected by pH or degradation products. The migration rate of the cells was also not inhibited by the extracts. Grémare et al. (2018) reported a similar result and showed the absence of a cytotoxic effect when using media extracts from printed PLA scaffolds [74]. These results confirm our toxicity and adhesion studies, which showed that TPU, PLA, and their combinations are non-toxic and biocompatible in cell cultures.

### 3.4. Cell Attachment on the 3D Printed Scaffold

The attachment or adhesion of cells on the material is the basis for subsequent cell development and differentiation and an important indicator of the biocompatibility of the material. When cells are incubated with toxic biological materials, changes in morphology, proliferation, and adhesion provide valid data for assessing biocompatibility. Cell adhesion to our 3D-printed scaffolds was evaluated using an attachment assay in which cells were seeded directly onto the 3D-printed scaffolds. Within seven days, the cells not only adhered to and dispersed on the solid surfaces of the porous scaffold, but also migrated into the interior of the porous scaffolds, as seen in the fluorescence images in Figure 5. The epithelial cells were scattered on the TPU scaffold and its composite, while they were present in denser populations in the porous region of the PLA scaffold. This result is consistent with a study by Jing et al. (2014), which showed that the cells accumulated and proliferated on the surface and in the porous areas [28]. This result could prove that the 3D printed TPU/PLA mixture has a positive biological effect on cell attachment and growth.

### 3.5. Macroscopic Evaluation, Cell Attachment, and Morphology under SEM

Recovery from anaesthesia after surgery was rapid and without problems. All animals survived throughout the study and showed no remarkable morbidity or behavioural changes. The dermal incisions and implantation sites were inspected daily and appeared clean and neat, with no signs of abscesses or severe inflammation.

The explanted scaffolds were retrieved and examined using SEM for attachment and migration of cells in vivo. Figure 6 presents the attachment and proliferation of cells on the scaffold surfaces at different time points. In the first week post-implantation, multiple single cells were attached on the surfaces of all scaffolds; especially on the TPU scaffolds the cells were large and flat-shaped. In contrast, on the TPU/PLA and PLA scaffolds, many mixed round and flattened cell shapes were observed. The presence of lamellipodia and filopodia indicates good cell adhesion, migration, and proliferation of the cells on the surface of the scaffolds. At week 4, the cells differentiated and proliferated, formed cell colonies, and spread over the scaffold surface, indicating stronger adhesion to the material. After 8 weeks, more fibrous tissue was seen on all scaffolds; thick on TPU/PLA, thin on PLA, and less fibrous on TPU. In general, all scaffolds showed good biocompatibility in the skin tissue, but the type of attached cells could not be identified with SEM.

Subcutaneous implantation in small animal models is commonly used to investigate the biocompatibility of tissue-engineered scaffolds for their immunological reactivity and recellularisation potential in a preclinical setting [75,76,77,78]. The results of implantation allow for observation of an inflammatory response, extracellular matrix recellularisation, and validation of angiogenesis [44,45,79,80,81]. Cell adhesion plays a role in signalling which controls cell differentiation, cell cycle, cell migration, and cell survival [82]. The SEM results showed good adhesion, migration, and proliferation of host cells on the surface of the scaffolds. The presence of multiple lamellipodia and filopodia suggests cell–matrix interactions driven by the surface topography of the scaffold [83]. Different degrees of adhesion evident in the number of cells on the surface of the different scaffold types during the first week provided evidence that initial cell adhesion may be determined by the surface topography of the scaffolds [84]. However, this parameter was not investigated in this study.

### 3.6. Inflammatory Response Following Implantation

H&E staining of composite scaffolds at 1, 4, and 8 weeks post-implantation was used to assess scaffold biocompatibility and inflammatory cell infiltration at the implant site. Figure 7a shows general views of histological cross-sections of the wound area representing the TPU, TPU/PLA, and PLA scaffolds at weeks 1, 4, and 8, respectively. The inflammatory response was quantified by counting the number of inflammatory cells in the skin area adjacent to the scaffold. One-way ANOVA with Tukey’s multiple comparison test was used to compare between groups, as shown in Figure 7b. It was visible that there was a significant difference (*p*-values < 0.0001) in the number of inflammatory cells between all groups at all time points compared to the naïve group. In week 1, the number of inflammatory cells was highest in all experimental groups and decreased significantly in weeks 4 and 8.

The tissue response to a foreign body is similar to the typical response to tissue injury, except for more prolonged proliferation and remodelling phases [44]. Implanted biomaterials are known to trigger acute inflammatory responses of variable severity, and this difference has been associated with the composition of the scaffold material [85]. Our findings in this section verified the biocompatibility results of the TPU, PLA, and the blend in a cell culture study. Only a mild to moderate inflammatory response was observed following subcutaneous implantation of the scaffolds in the rat model. The inflammatory response did not continue for more than eight weeks, and healing was observed. This finding is in accordance with the report by Jaiswal et al. (2013), who implanted an electrospun PLA scaffold subcutaneously onto the dorsum of the rats. Similarly, Chocarro-Wrona et al. (2021) showed nontoxicity and biocompatibility of a 3D bio-printed TPU elastomer in immunocompetent mice [86].

### 3.7. Vascularisation in the Scaffold Surroundings Post-Implantation

Vascularisation is one of the critical components for tissue regeneration, which has long been a problem in tissue engineering, especially the trachea. Vascularisation on the scaffolds was analysed via semi-quantitative analysis of H&E staining. Compared with the naïve group, vascularisation at week 1 showed minimal changes in both the TPU and PLA groups, while it was significantly higher in the sham group. However, at week 4, a significant increase was seen in PLA, followed by the TPU/PLA blend and TPU. At week 8, vascularisation was almost back to baseline as shown in Figure 8.

The 3D-printed porous scaffolds were developed in the current study as an option to facilitate cell integration. It is considered that the appropriate porosity of the scaffold mimics the extracellular matrix to provide an ideal environment for cell adhesion, migration, and proliferation through the scaffold. The pores and porosity within the scaffolds provide an additional surface area for cell development and allow for the diffusion of nutrients, oxygen, and metabolites through the scaffolds. The scaffolds we prepared had a pore size of 120 to 150 µm and an average porosity of 30 to 40 µm with interconnections between the pores. Research by Xiao et al. (2017) showed that endothelial cells tend to cluster around the pore junctions of porous scaffolds, where the cells migrate and initiate vascularisation [87]. Furthermore, Jung et al. (2016) reported fibrotic connective tissue with neovascularisation in the porous inner structure of a 3D-printed TPU scaffold four weeks after transplantation into the trachea of rabbits [88]. Gao et al. (2019) reported in vivo transplantation of a chondrocyte-populated porous 3D-printed PLA scaffold with additional prevascularisation into the tracheal defect of a rabbit. The autologous chondrocytes seeded on the scaffolds survived well with adequate blood supply during implantation and had the ability to synthesise and secrete matrix, which facilitated the maturation of hyaline cartilage tissue [69]. Overall, the pores and porosity of the scaffold facilitate the angiogenesis process, improving the properties of the scaffold.

The 3D printing technology has advanced rapidly over the years. A newer method, extrusion-based 3D bioprinting, has been used to create artificial structures from different materials and in different designs. This technique has shown great promise in producing heterogeneous constructs using a variety of cell types and biocompatible polymers to create cellular tracheal structures that mimic the biological and physiological function of the natural trachea [89,90]. Although 3D-printed products excellently mimic physiological properties, biomedical devices made with this technology are static and not intended for use under dynamic conditions. Native tissues in the living body have a microenvironment that promotes their development and controls some of their biological processes. Bioactive materials should be able to adapt to the changing environment and also meet stringent biodegradability and biocompatibility standards for better performance [91]. The latest manufacturing method is 4D bioprinting, an improved version of 3D printing that offers higher quality, precision, accuracy, and performance and can produce any intricate object from a variety of materials [92]. TPU and PLA may be among the materials that can be used to exploit this technology in future research.

## 4. Conclusions

In this study, the degradability, cell attachment, and effects of TPU, PLA, and blend extracts were investigated in cell culture studies, while the biocompatibility of the materials was evaluated both in vitro and in vivo. From the SEM results, using predefined parameters, 3D printing technology can be used to produce a customised scaffold, which is crucial for implantation. Our data from the in vitro studies showed that the mixed material and its extracts were biocompatible with the cells, and the pH of the extracts had no effect on cell proliferation and migration. In addition, the biodegradation rate was relatively slow, making it suitable for regenerative therapies such as tracheal replacement. The in vivo study confirmed the results and indicated the potential use of porous TPU/PLA scaffolds to facilitate cell adhesion, migration, and proliferation and to promote angiogenesis in host cells. Furthermore, the ability to customise the design and parameters of polymeric scaffolds using 3D printing technologies facilitates the integration of host cells into the matrix and promotes tissue regeneration. The TPU and PLA blend scaffolds fabricated in the present study using a 3D printing technology are likely to be promising for the development of synthetic tracheas in tissue engineering.

## Figures and Tables

**Figure 1 bioengineering-10-00394-f001:**
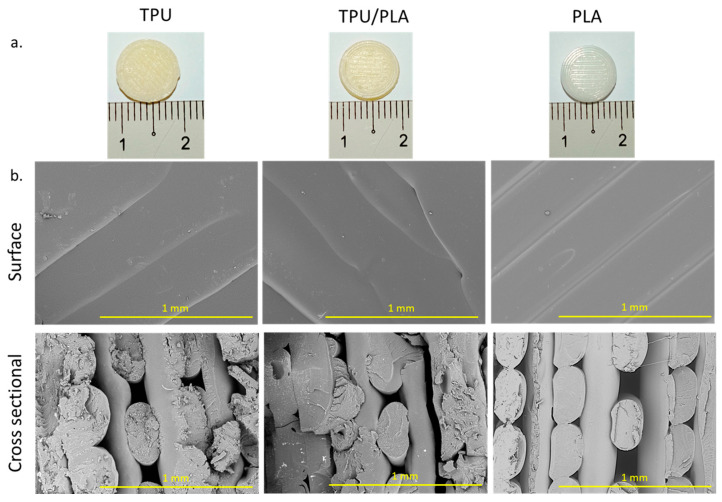
(**a**) Photographic images of the constructed disc-shaped scaffold. Images of TPU, TPU/PLA, and PLA scaffolds from the top view. Scale bar: 1 cm. (**b**) SEM images of the disc-shaped scaffolds from the surface and cross-sectional views of all scaffolds. Scale bar: 1 mm.

**Figure 2 bioengineering-10-00394-f002:**
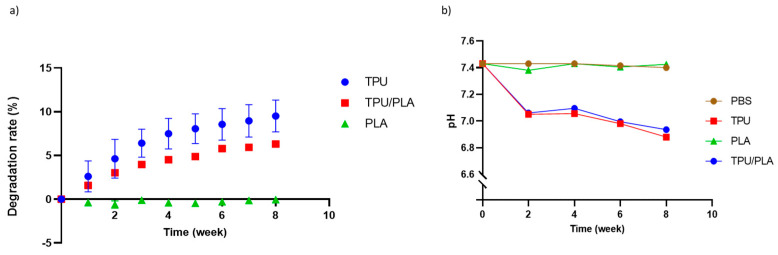
(**a**) The degradation rate of porous 3D-printed scaffolds. The TPU scaffold showed the highest degradation rate compared to other scaffolds. Moreover, the longer the time period, the higher the degradation rate. (**b**) pH analysis of the 3D printed scaffold extracts. The PLA extract showed no significant difference compared to PBS. However, in the first two weeks, a significant decrease in pH was observed in both the TPU and TPU/PLA extracts, which gradually reduced to below pH 7 after eight weeks.

**Figure 3 bioengineering-10-00394-f003:**
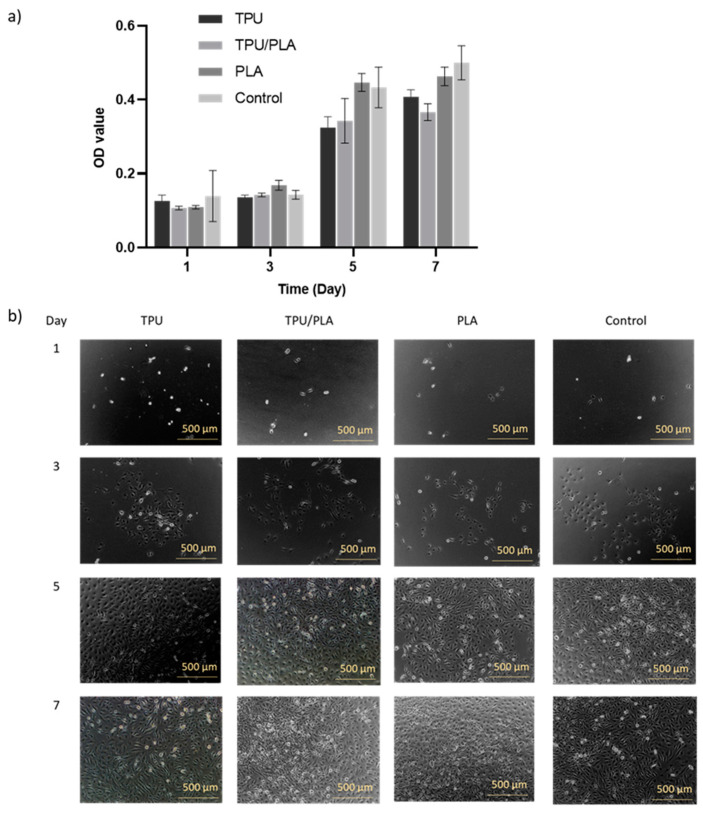
(**a**) Indirect proliferation assay using scaffold extracts. The *X*-axis represents the incubation time in days, while the *Y*-axis represents the optical density values of the BEAS-2B cells. In general, the proliferation of BEAS-2B cells was almost consistent from day 1 to day 3 but increased dramatically on day 5 and slowed down on day 7 for all types of scaffolds. Significant proliferation activities were noted between all types of scaffolds over time. (**b**) The proliferation of BEAS-2B cells in immersion media of all scaffold types. Phase contrast micrographs showing the percentage of confluence of proliferating BEAS-2B cells at days 1, 3, 5, and 7. It was noted that the confluences of the cells in all groups were almost similar. Scale bar: 500 µm.

**Figure 4 bioengineering-10-00394-f004:**
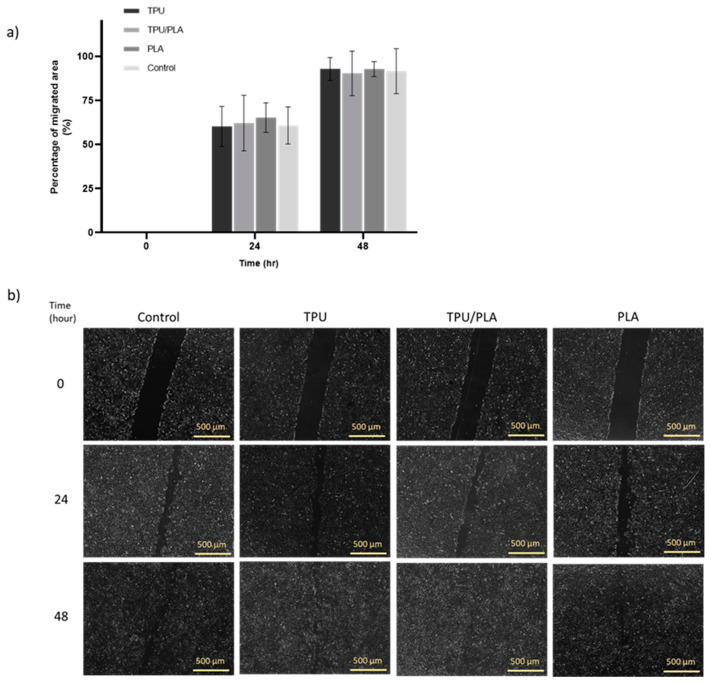
(**a**) BEAS-2B cell scratch analysis. (**b**) The effects of extracts from the 3D printed scaffolds on the migration of BEAS-2B cells. The scratches were made and the cells were treated with different scaffold extracts. Images of the wound area were taken 0, 24, and 48 h after treatment and measured using ImageJ analysis software. Scale bar: 500 µm.

**Figure 5 bioengineering-10-00394-f005:**
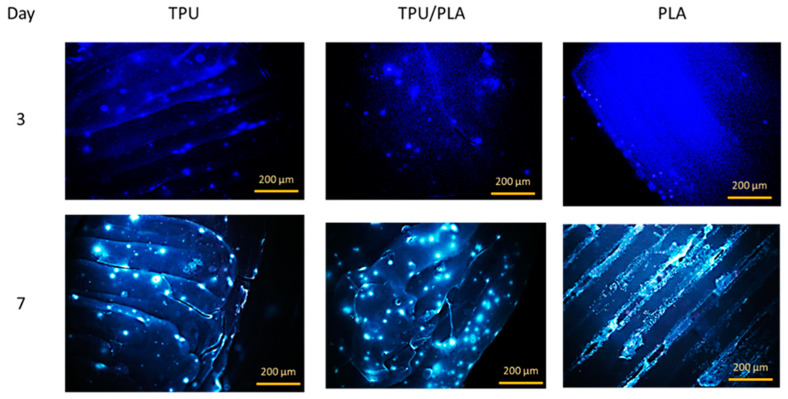
Direct cell attachment assay using Hoechst 33342. The fluorescence image shows that cells had attached to all three scaffold surfaces on day 3. On day 7, more cells were visible, especially between the PLA filaments. The cells appeared slightly out of focus as they were inside the scaffold folds. Scale bar: 200 µm.

**Figure 6 bioengineering-10-00394-f006:**
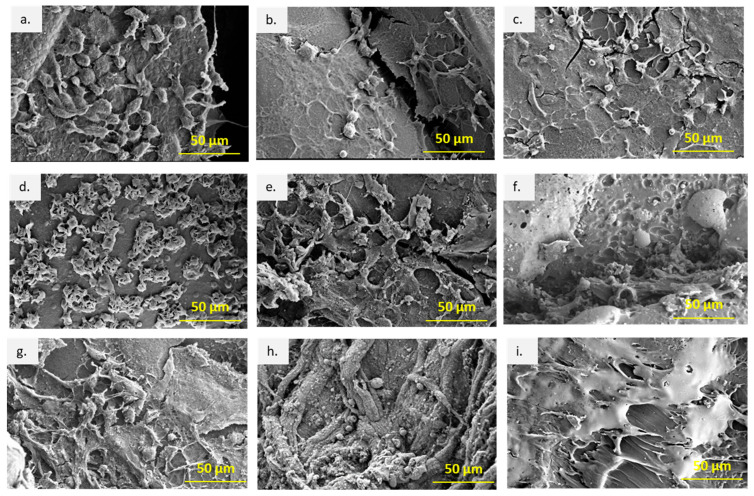
SEM images of the cells and tissues following subcutaneous implantation at different time points. (**a**) TPU at week 1, (**b**) TPU/PLA at week 1, (**c**) PLA at week 1, (**d**) TPU at week 4, (**e**) TPU/PLA at week 4, (**f**) PLA at week 4, (**g**) TPU at week 8, (**h**) TPU/PLA at week 8, and (**i**) PLA at week 8. Scale bar 50 µm.

**Figure 7 bioengineering-10-00394-f007:**
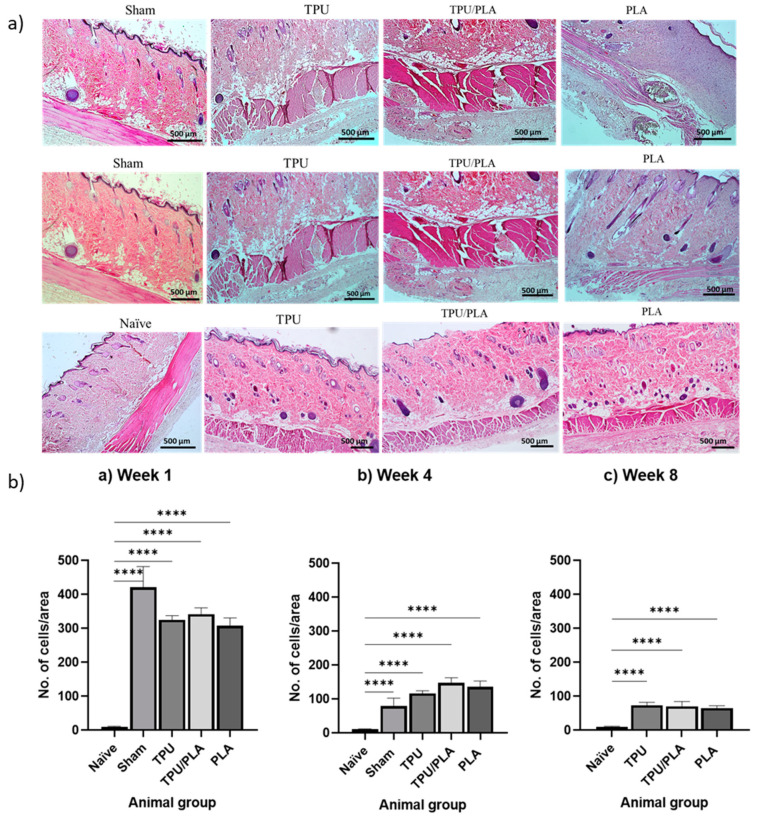
(**a**) Photomicrograph of tissue adjacent to the implanted scaffold showing inflammatory reaction in rat skin. Histological sections of a representative subcutaneous area were stained with H&E at 4× magnification. (**b**) Semi-quantitative scoring of the number of inflammatory cells in rats. The number of inflammatory cells at (a) week 1, (b) week 4, and (c) week 8. The *X*-axis represents the animal groups, while the *Y*-axis indicates the number of inflammatory cells. At week 1, inflammatory responses were significantly different and highest in all experimental groups compared to the naïve group, significantly reduced at week 4, and gradually decreased at week 8, with mild inflammation present. One-way ANOVA with Tukey’s multiple comparison test was used for comparison between groups. Data are presented as mean ± SD, n = 2 animals per group, **** *p* < 0.0001.

**Figure 8 bioengineering-10-00394-f008:**
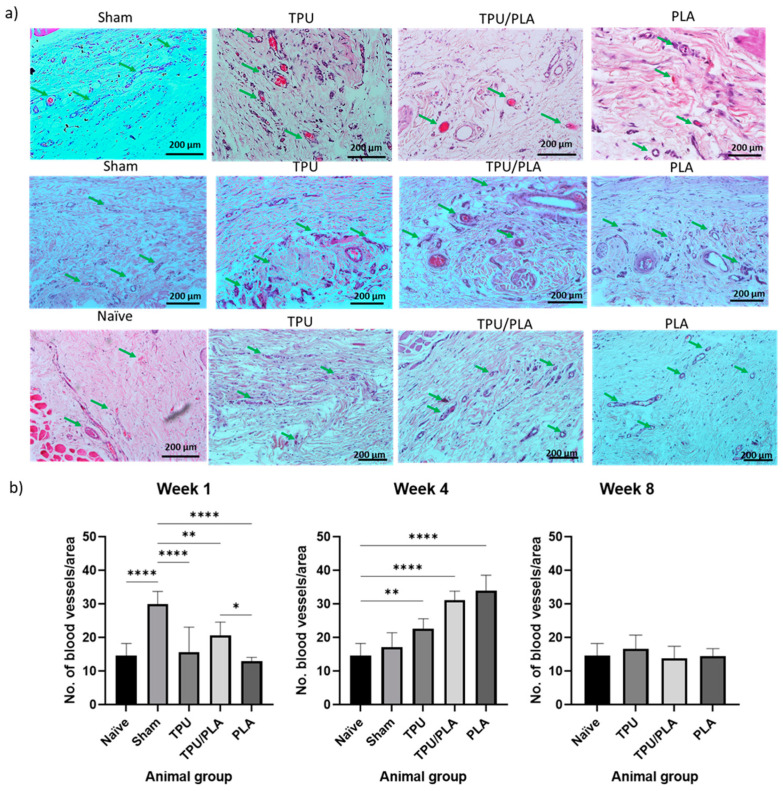
(**a**) Vascularisation of the tissue adjacent to the implanted scaffold at week 1. H&E staining shows blood vessels in the tissue adjacent to the implanted scaffold. The green arrow indicates the blood vessels. The highest number of blood vessels was observed in the sham and PLA groups. (**b**) Semi-quantitative analysis of vascularisation at different time points. Week 1, week 4, and week 8. The *X*-axis represents the grouping of animals, and the *Y*-axis represents the number of blood vessels per area. Compared to the naïve group, vascularisation at week 1 showed minimal changes in both the TPU and PLA groups, while it was significantly the highest in the sham group. However, a significant increase was observed in week 4, which was highest in PLA, followed by the TPU/PLA blend, and TPU. Finally, vascularisation was almost back to baseline at week 8. One-way ANOVA with Tukey’s multiple comparison tests was used for comparison between groups. Data are presented as mean ± SD, n = 2 animals per group, * *p* < 0.05, ** *p* < 0.01 **** *p* < 0.0001.

**Table 1 bioengineering-10-00394-t001:** Mean fibre diameter, pore size, and porosity of the scaffolds.

Polymer Type	Mean Fibre Diameter Size (µm)	Mean Pore Size (µm)	Mean Porosity (%)
TPU	205.00 ± 82.25	133.28 ± 38.22	31.3 ± 12.4
TPU/PLA	183.80 ± 43.25	129.60 ±43.08	27.3 ± 5.8
PLA	278.40 ± 61.09	122.64 ± 45.16	38.7 ± 22.3

No significant difference between all parameters (*p*-value > 0.5).

## Data Availability

Not applicable.

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
