# Peer review of "Investigation of the In Vitro and In Vivo Biocompatibility of a Three-Dimensional Printed Thermoplastic Polyurethane/Polylactic Acid Blend for the Development of Tracheal Scaffolds"

_bioengineering, 2023, doi:10.3390/bioengineering10040394_

Round 1

Reviewer 1 Report

The abstract is written very briefly and superficially. The abstract should appeal to the reader. Novelty should be presented transparently. Research achievements and results should be mentioned quantitatively.

Be careful in choosing keywords. For example, the term "inflammatory response" is used, but it is not mentioned at all in the abstract.

The number of references used is high, especially in the introduction section. Remove older sources. It also seems that the format used is inconsistent with the journal's format. In addition, it is suggested to use related papers in this field, especially PLA-TPU FDM printing in the introduction (“3D printing of PLA-TPU with different component ratios: Fracture toughness, mechanical properties, and morphology” “Development of Pure Poly Vinyl Chloride (PVC) with Excellent 3D Printability and Macroand MicroStructural Properties” and “A New Strategy for Achieving Shape Memory Effects in 4D Printed Two-Layer Composite Structures.

It is suggested that the research method be more concise and provide a better classification for the investigated cases.

Explain the printing parameters used. It is necessary to mention the basic printing parameters such as velocity, layer thickness, temperature, nozzle diameters, etc.

It is not customary to use raw images in research articles. The SEM images should be presented in a better way and the scalebar should be removed.

The numbering of subsections of the results has been forgotten.

The number of the figure used is high, and some of them should be merged or deleted.

In conclusion, for a better understanding, refer to quantitative data.

More analysis and discussion should be done in each result section. Most results are reported only.

Author Response

Dear reviewer,

Thank you

Reviewer 2 Report

Respected Authors,

It is a pleasure to accept the task of reviewing your Manuscript entitled “Investigation of the in vitro and in vivo biocompatibility of three-dimensional printed thermoplastic polyurethane/polylactic acid blend for development of tracheal scaffold”. In this article, the authors developed PLA/TPU-based scaffolds by using 3D printing technology and investigated their biological and mechanical properties.

Overall, the research article still has some shortcomings, which need to be addressed before possible publication in bioengineering journal.

Please find the attached annotated file to see my comments.

Once the authors agree to take my points into account, I will be happy to accept the manuscript. To summarize this, I would like to say Bioengineering is a competitive journal, which published high-quality research articles related to tissue engineering. Based on my comments, the recommendation is Major Revision.

Author Response

Dear reviewer,

Thank you.

Round 2

Reviewer 1 Report

Accept.

Reviewer 2 Report

Accept